# Cell-free tumour DNA analysis detects copy number alterations in gastro-oesophageal cancer patients

Karin Wallander[1,2], Jesper Eisfeldt[1,2], Mats Lindblad[3,4], Daniel Nilsson[1,2], Kenny Billiau[5], Hassan Foroughi[5], Magnus Nordenskjöld[1], Agne Liedén[1,2⊙], Emma Tham[1,2⊙]*

**1** Department of Molecular Medicine and Surgery, Karolinska Institutet, Stockholm, Sweden, **2** Department of Clinical Genetics, Karolinska University Hospital, Stockholm, Sweden, **3** Department of Clinical Science, Intervention and Technology, Karolinska Institutet, Stockholm, Sweden, **4** Department of Upper Abdominal Diseases, Karolinska University Hospital, Stockholm, Sweden, **5** Department of Microbiology, Tumor and Cell Biology, Karolinska Institutet, Stockholm, Sweden

⊙ These authors contributed equally to this work.
* emma.tham@ki.se

## Abstract

**Data Availability Statement:** All relevant data are within the paper and its Supporting information files.

### Background

Analysis of cell-free tumour DNA, a liquid biopsy, is a promising biomarker for cancer. We have performed a proof-of principle study to test the applicability in the clinical setting, analysing copy number alterations (CNAs) in plasma and tumour tissue from 44 patients with gastro-oesophageal cancer.

### Methods

DNA was isolated from blood plasma and a tissue sample from each patient. Array-CGH was applied to the tissue DNA. The cell-free plasma DNA was sequenced by low-coverage whole-genome sequencing using a clinical pipeline for non-invasive prenatal testing. WISECONDOR and ichorCNA, two bioinformatic tools, were used to process the output data and were compared to each other.

### Results

Cancer-associated CNAs could be seen in 59% (26/44) of the tissue biopsies. In the plasma samples, a targeted approach analysing 61 regions of special interest in gastro-oesophageal cancer detected cancer-associated CNAs with a z-score >5 in 11 patients. Broadening the analysis to a whole-genome view, 17/44 patients (39%) had cancer-associated CNAs using WISECONDOR and 13 (30%) using ichorCNA. Of the 26 patients with tissue-verified cancer-associated CNAs, 14 (54%) had corresponding CNAs in plasma. Potentially clinically actionable amplifications overlapping the genes *VEGFA*, *EGFR* and *FGFR2* were detected in the plasma from three patients.

**Funding:** KW, ET and MN are supported by grants from Region Stockholm (combined clinical residency and PhD training program: K2891-2016 and ALF-project: grant number 581046). CA17118 (European Cooperation in Science and Technology, www.cost.eu) has provided networking support to ET.

**Competing interests:** The authors have declared that no competing interests exist.

**Abbreviations:** Array-CGH, array- comparative genomic hybridisation; CIN, chromosomal instability; CNA, copy number alteration; ctDNA, cell-free tumour DNA; NIPT, non-invasive prenatal test; WISECONDOR, WIthin-SamplE COpy Number aberration DetectOR.

## Conclusions

We conclude that low-coverage whole-genome sequencing without prior knowledge of the tumour alterations could become a useful tool for cell-free tumour DNA analysis of total CNAs in plasma from patients with gastro-oesophageal cancer.

## Introduction

The stomach and the oesophagus are the fifth and seventh most common cancer locations worldwide but cancers in these organs are the third and sixth most common causes of cancer death. Symptoms of gastro-oesophageal cancer are often diffuse and develop slowly why the diagnosis frequently is set at a late stage [1]. No reliable diagnostic clinical biomarker is available [2].

Previous classifications of gastro-oesophageal cancer have been based mainly on morphology, but now molecular classifications are gaining ground. The Cancer Genome Atlas Network has suggested four gastric cancer subtypes: Epstein–Barr virus positive tumours, microsatellite instable tumours, genomically stable tumours (with a high proportion of diffuse histological subtype), and tumours with chromosomal instability (CIN) [3]. Other classifications have also been suggested [4, 5]. In the oesophagus, both squamous cell carcinoma and adenocarcinoma can occur. Oesophageal adenocarcinoma share many clinical and epidemiological characteristics with gastric adenocarcinoma [3] and there is also similarity between the genetic aberrations in squamous cell cancer and adenocarcinoma in the oesophagus [6].

Thanks to the increase in molecular tumour analyses, targeted therapy is being introduced in metastatic gastro-oesophageal cancer. For instance, ERBB2-inhibitors have been recommended for use in metastatic gastro-oesophageal cancer [7] and there are possibly many other molecular subgroups that can guide treatment [8–10]. Gastro-oesophageal cancer have no hotspot mutations but copy number alterations (CNAs) are common and therefore, CNA analysis could potentially identify at least half of all gastro-oesophageal cancer [11].

An emerging biomarker in the field of cancer is liquid biopsy, including analysis of cell-free tumour DNA (ctDNA) in plasma from patients with cancer. Fragmented DNA is released into the bloodstream, both from cells within the blood system and from solid tissue cells, and the fraction originating from tumour cells consequently harbours the same genetic aberrations as the tumour [12]. It has been shown that analysis of ctDNA can detect genetic aberrations from many different cancer types, including gastric cancer [13]. ctDNA reflects the total mutational burden of the tumour cells and may thus be a valuable complement to tissue biopsies, reducing the problems of heterogeneity in tissue biopsies in gastro-oesophageal cancer and being more accessible [14].

To investigate the potential of ctDNA analysis to detect and characterize gastro-oesophageal cancer CNAs, we have performed a proof of concept study in which we have compared CNAs detected by tumour tissue sample array-CGH (comparative genomic hybridization) to plasma ctDNA copy number analysis. We tested two different bioinformatic solutions for this analysis, WISECONDOR [15] and ichorCNA [16].

## Methods

### Participants

All patients with newly diagnosed gastric adenocarcinoma referred to the Department of Gastric Upper Abdominal Diseases at Karolinska University Hospital, Stockholm, Sweden, from

June 2016 to May 2018 were asked to participate. From March 2017 to September 2018 newly diagnosed oesophageal cancer patients were also asked for inclusion. This study was approved by the Regional Ethical Review Board in Stockholm (reg number 2016/2-31/1) and all participants provided written informed consent for participation in the study and for publication of the results.

In this pilot substudy, only participants with an available tumour tissue analysis and at least one plasma sample drawn in connection to the time of diagnosis were included. The time of other treatments, such as chemotherapy, radiotherapy and surgery, were noted for each participant in relation to the time of plasma and tissue sample collection.

## Tissue array-CGH analysis

Tumour tissue samples were either extracted during gastroscopy or surgery and frozen within one day. Isolation of DNA was carried out using the EZ1 DNA Tissue Kit (Qiagen) according to the manufacturer's protocol. The microarray analysis was performed using a 180K oligonucleotide array with evenly distributed whole-genome coverage (AMAID 031035, Oxford Gene Technology) as previously described [17].

The array results were analysed using the Cytosure Interpret Software version 4.10.41 (Oxford Gene Technology, Begbroke, UK). CNAs were considered cancer-associated if they did not overlap with previously described CNAs in our internal database (~8000 samples) or published data sets [18–20] and had a log2 ratio that did not match a germline variant. We classified a tissue sample as chromosomally instable (CIN) if there were cancer-associated gains or losses on 10 or more chromosomes. There is no clear cut-off defining a general amplification in any gene and we defined the tissue array-CGH gains as amplifications when the log2 ratio indicated 5 or more copies [21].

In one selected case, a finding on array-CGH was verified using a targeted gene panel of 370 genes. This sample was prepared for sequencing using the KAPA library preparation kit (Roche Sequencing, CA, USA) and the Twist hybridization protocol (Twist Bioscience, CA, USA) and were sequenced on a NovaSeq 6000 system (Illumina, CA, USA).

## Plasma ctDNA analysis

The samples were processed using the manual 16-plex VeriSeq workflow, which is routinely used for clinical non-invasive prenatal test (NIPT) samples at the Department of Clinical Genetics. Briefly, blood samples were collected in cell-free DNA blood collection tube (STRECK, La Vista, USA). The samples were centrifuged 1600g for 10 minutes at room temperature to separate the plasma from the blood cells. Plasma was transferred to microcentrifuge tubes and centrifuged at 16,000g for 10 minutes at 4˚C and the supernatant was stored at -80˚C. All plasma samples were separated within 5 days of the blood draw. Cell-free DNA was extracted with the QIAamp DNA Blood Mini Kit (Qiagen, Hilden, Germany) from 1 ml of plasma and converted to libraries for sequencing using the TruSeq Nano DNA LT Sample Prep Kit (Illumina, San Diego, USA), with 13 cycles of amplification. Whole-genome low coverage (36 bp single-end) sequencing was performed on an Illumina HiSeq 2500 with an average of 23M reads per sample (range 14-49M).

Sequence reads were aligned to the reference genome (GRCh37/hg19) using BWA aln [22], deduplicated with Picard tools (http://broadinstitute.github.io/picard/), and converted and analyzed using the WISECONDOR (WIthin-SamplE COpy Number aberration DetectOR)) program [15]. The software was accessed (https://github.com/VUmcCGP/wisecondor) in December 2017. 414 NIPT samples without any known foetal aberrations were used as a reference set. As a first step, the performance of a targeted approach for 61 target regions using

WISECONDOR was evaluated. The target regions were selected as recurrently harbouring gastro-oesophageal cancer CNAs, according to the Cancer Genome Atlas Network [3] (S1 Table). The z-score limit was set to 3.0 and a 500 kB bin size was used for the genes and for larger chromosomal regions we combined the z-scores of all bins to a median Z-score for the region.

Thereafter, a whole-genome analysis was performed using two different software. With WISECONDOR, a sliding window method was used to identify the most significant sequence of bins (Stouffer's z-score). A bin size of 500 kb, and a minimum of 25 reference bins (all mapping on other chromosomes than the target bin) were used. CNA calls were made if they had a z-score of at least 4.95 and a minimal effect size of 1.5% (i.e. approximately a 1.5% difference in target bin sequencing coverage). We tested larger bin sizes in WISECONDOR (1, 5 and 15 Mb), to reduce the number of tests and therefore allow a lower z-score threshold. However, this did not result in any additional verified cancer CNAs being detected.

Sequencing data was also analysed using the ichorCNA algorithm as previously described [16]. The software was accessed at (https://github.com/broadinstitute/ichorCNA) in February 2018. The ctDNA fraction is defined as the ratio of DNA derived from the tumour cells to the total cell free DNA. The same 414 NIPT samples as in WISECONDOR were used as a reference set ("panel of normals"). A bin size of 500kb and default settings without subclonal analysis were used in accordance with the instructions for low ctDNA fraction samples (https://github.com/broadinstitute/ichorCNA/wiki/Parameter-tuning-and-settings). Only calls with an effect size of 1.5% (converted from the reported segmental log2 ratio) or higher were included for further analysis, in accordance with the WISECONDOR analyses.

Recurrent calls from segmental duplication regions, variable centromere regions and likely germline variants together with calls present in the reference set were filtered out from both the WISECONDOR and ichorCNA data sets. Chromosome X and Y were not included in the analysis. In addition, calls from chromosome 19, a GC-rich chromosome with known normalization problems, were filtered out [15, 16]. After this, the remaining likely cancer-associated CNAs were classified as verified if they were detected by array-CGH in the paired tissue or unverified if they were not. Amplification status was determined in relation to the ctDNA fraction, when ichorCNA provided an estimate, and the ratio effect size/ctDNA fraction was used in those samples. A ratio above 1.5, indicating a copy number status of at least 5 in the tumour cells was considered an amplification. In the samples where no ctDNA fraction was calculated by ichorCNA, an effect size above 4.5%, corresponding to a copy number status of 5 if the ctDNA fraction was 3%, was considered indication of an amplification. In both the plasma and the tissue analyses, eight potentially clinically actionable gene amplifications, according to other studies [2, 23–27], were noted.

## Reference set and positive control samples

De-identified data from a set of 414 and NIPT samples without any known foetal aneuploidies were used as reference samples for the WISECONDOR and ichorCNA analysis as described above. All the WISECONDOR and ichorCNA calls in the reference set are listed in S2 Table. In addition, sequencing data from 15 verified foetal aneuploidies were used as positive controls in the analysis set up. The foetal fraction in these samples was estimated using SeqFF [28]. All of the foetal aneuploidies could be detected by WISECONDOR and ichorCNA (S3 Table).

We also evaluated the reference set for samples with high variation that could potentially decrease the sensitivity. Comparing the bin coverage difference (mean absolute error) to the number of calls for the samples we saw no clear correlation for the WISECONDOR data (S1a Fig). However, the ichorCNA data showed a tendency towards an increased number of calls

(>10) if the mean absolute error was >2.5% for the sample (S1b Fig). Therefore, the 69 reference samples with a mean absolute error >2.5% were excluded from the reference set. The use of the adjusted reference set did not affect the number of verified cancer CNAs or the ctDNA fraction estimations.

## Results

81 patients were included in the study during a two-year period. Out of those 81, fresh frozen tumour tissue biopsies and plasma samples taken around the time of diagnosis were available in 44 (Fig 1). The demographic characteristics of the included patients are shown in Table 1. The median age was 70 years and 30% were women. The tumour stages ranged between 0 and IV, with the most common stage being IIIC.

### CNAs detected in genomic DNA from tumour tissue

Cancer-associated CNAs (listed in S4 Table) could be seen in 59% (26/44) of the fresh frozen tumour biopsies (Fig 1 and Table 2). A total of 60 amplifications in 14 patients were detected in the tumour tissue array-CGH analyses. In patients P03 and P35, potentially clinically actionable amplifications were detected (Fig 2). In sample P03, there was an amplification of the *EGFR* gene and also an amplification on chromosome 17 (genome position 38–40 Mbps) adjacent to, but not including the *ERBB2* gene. Analysis using a specific targeted gene panel confirmed that the amplification did not encompass *ERBB2* (S6 Table). In P35 there were amplifications of the genes *VEGFA* and *ERBB2*.

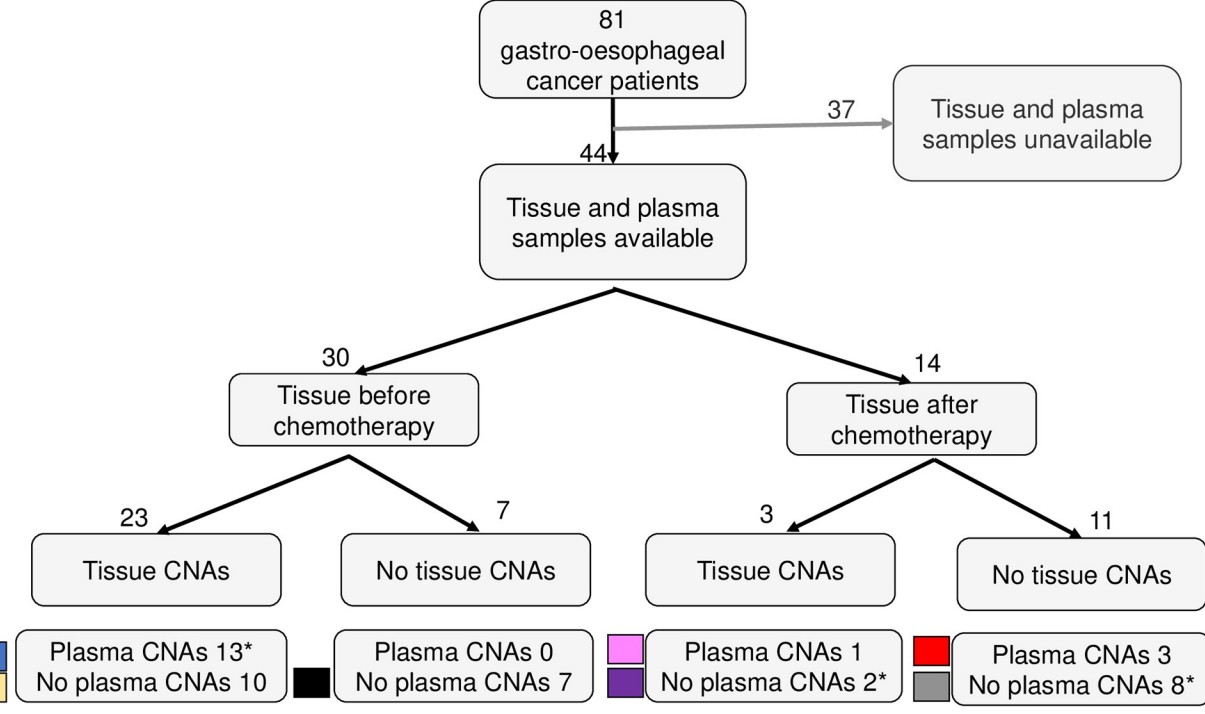

**Fig 1. Gastro-oesophageal cancer patients, collected samples and overview of the study results.** Flowchart including all the patients divided into groups based on treatment and detected cancer-associated copy number alterations (CNAs) in tumour tissue biopsy and/or plasma. *One plasma sample in each of these three groups (P05, P19, P35) was drawn after chemotherapy. arr; array-CGH (tumour tissue). WC; WISECONDOR (plasma). ichor; ichorCNA (plasma).

**Table 1. Demography.** The clinical data from the patients included in the study.

| | Individual | Cancer type | Sex | Age | Tissue sample before chemo | Verified metastases? (a) | Stage at time of plasma sample | Stage at time of tissue sample | Tumour size (cm) |
|---|---|---|---|---|---|---|---|---|---|
| TP | P01 | GC diffuse | W | 56 | yes | no | IIIC(c) | IIIC(c) | 11 |
| TP | P02 | GC | M | 77 | yes | yes | IV(c) | IV | NA |
| TP | P03 | GC | M | 64 | yes | no | IIIB(c) | IIIB(c) | 2.5 |
| TP | P04 | GC | M | 50 | yes | no | IIIC | IIIC | 12 |
| TP | P05 | GC | M | 60 | yes(b) | no | IIIC | IIB(c) | 11 |
| TP | P06 | GC | M | 65 | yes | no | IIIA(c) | IIIA(c) | 5 |
| P | P07 | Eso squamous | M | 64 | no | no | IIIA(c) | 0 | 0 |
| TP | P08 | GC diffuse | M | 68 | yes | yes | IV(c) | IV(c) | NA |
| TP | P09 | GC diffuse | W | 83 | yes | yes | IIIA(c) | IIIA(c) | NA |
| TP | P10 | Eso squamous | W | 76 | yes | no | IIIC | IIIC | 9 |
| TP | P11 | GC | M | 84 | yes | no | IA | IA | 10 |
| TP | P12 | GC | M | 79 | yes | no | IIIC | IIIC | 5.5 |
| P | P13 | GC | M | 68 | no | no | IIIA(c) | I-IIB | 14 |
| TP | P14 | GC diffuse | M | 75 | yes | no | IIA(c) | IIA(c) | 5 |
| T | P15 | GC | M | 76 | yes | no | IIB(c) | IIIC | 5 |
| TP | P16 | GC | M | 72 | no | no | IIB(c) | IIIA | 2 |
| TP | P17 | Eso | M | 78 | yes | no | IIIC | IIIC | 6 |
| N | P18 | GC diffuse | M | 43 | no | yes | IV(c) | 0 | 13 |
| N | P19 | GC diffuse | W | 71 | no(b) | no | IIIC | IIIC | 20 |
| T | P20 | GC | M | 85 | yes | no | IIIC | IIIC | 4 |
| T | P21 | Eso | M | 48 | no | no | IIIB(c) | IIIC | 4.5 |
| T | P22 | GC diffuse | M | 77 | yes | no | IIB | IIB | 3.5 |
| T | P23 | GC diffuse | W | 75 | yes | no | IIIC | IIIC | 3.5 |
| N | P24 | GC | M | 75 | no | no | IIA(c) | IIIA | 3 |
| N | P25 | GC diffuse | M | 71 | yes | no | IIIC | IIIC | 16 |
| T | P26 | GC | M | 81 | yes | no | IIB(c) | IIB | NA |
| N | P27 | GC | M | 79 | yes | no | IB | IB | 1.5 |
| T | P28 | GC | M | 79 | yes | no | IIB | IIB | 2.5 |
| T | P29 | Eso | M | 59 | yes | no | IIIB(c) | IIIB | 2 |
| N | P30 | GC | M | 33 | no | no | IIA(c) | IIA | <1 |
| T | P31 | GC | M | 78 | yes | no | IIB(c) | IIB(c) | 4 |
| N | P32 | GC diffuse | W | 61 | yes | yes | IV | IV | NA |
| N | P33 | GC diffuse | W | 56 | no | no | IB(c) | IB | 1 |
| N | P34 | GC | M | 55 | no | no | IIIA(c) | IB | 5 |
| T | P35 | GC | M | 66 | no(b) | no | IA | IA | 4 |
| N | P36 | GC diffuse | W | 68 | no | no | IB(c) | IB | <1 |
| N | P37 | GC | W | 75 | yes | no | IIIA(c) | IA | 4 |
| T | P38 | GC | W | 74 | yes | no | IIA(c) | IIIC | 7 |
| N | P39 | GC diffuse | W | 57 | yes | no | IA | IA | 5 |
| N | P40 | GC | M | 67 | yes | yes | IV | IV | NA |
| N | P41 | GC diffuse | M | 43 | no | no | IIA(c) | IIB | 2.5 |
| T | P42 | GC diffuse | W | 75 | yes | yes | IV(c) | IV(c) | NA |
| N | P43 | GC | M | 71 | yes | no | IIIA(c) | IIIA(c) | 3 |
| P | P44 | GC diffuse | W | 64 | no | no | IIIA(c) | IV | NA |

Group: Grey boxes indicate tissue sampled before chemotherapy. White boxes indicate tissue sampled after chemotherapy. T = CNAs detected in tissue only; P = CNAs detected in plasma only; TP: CNAs detected in both tissue and plasma, N = No CNAs detected in either tissue or plasma.

Chemo; chemotherapy. Eso: oesophageal cancer, adenocarcinoma, unless annotated "sq" for squamous. GC; gastric cancer. M; man W; woman. NA: no complete histopathology report available. Tumour size: Longest tumour measure at histopathology report [cm].

[a] Metastases diagnosed at any time before last sample included in study.

[b] Plasma sampled after chemotherapy.

[c] stage determined by clinical examination including radiology. In all other cases, stage was determined by histopathological analysis

**Table 2. Plasma and tissue findings.** CNAs detection in plasma and tissue for each patient.

| | Individual | CIN? | CNAs in tissue? | CNAs in plasma (WC)? | CNAs in plasma (ichor)? | ctDNA fraction ichor [%] | Hit in the targeted analysis? |
|---|---|---|---|---|---|---|---|
| TP | P01 | yes | yes | yes | yes | 8.4 | yes |
| TP | P02 | yes | yes | yes | yes | 4.3 | yes |
| TP | P03 | yes | yes | yes | yes | 4.2 | yes |
| TP | P04 | yes | yes | yes | yes | 5.2 | yes |
| TP | P05 | yes | yes | yes | no | 0 | yes |
| TP | P06 | yes | yes | yes | yes | 3.6 | yes |
| P | P07 | no | no | yes | yes | 2.8 | yes |
| TP | P08 | no | yes | yes | yes | 3.5 | yes |
| TP | P09 | yes | yes | yes | yes | 0 | yes |
| TP | P10 | yes | yes | yes | yes | 2.5 | no |
| TP | P11 | no | yes | yes | yes | 7.8 | no* |
| TP | P12 | no | yes | yes | yes | 4 | no |
| P | P13 | no | no | yes | yes | 0 | yes |
| TP | P14 | yes | yes | yes | no | 0 | no |
| T | P15 | yes | yes | no | no | 5.5 | no |
| TP | P16 | no | yes | yes | no | 0 | no |
| TP | P17 | yes | yes | yes | no | 0 | no |
| N | P18 | no | no | no | no | 0 | no |
| N | P19 | no | no | no | no | 0 | no |
| T | P20 | yes | yes | no | no | 0 | no |
| T | P21 | no | yes | no | no | 0 | no |
| T | P22 | no | yes | no | no | 0 | no |
| T | P23 | no | yes | no | no | 0 | no |
| N | P24 | no | no | no | no | 0 | no |
| N | P25 | no | no | no | no | 0 | no |
| T | P26 | no | yes | no | no | 0 | no |
| N | P27 | no | no | no | no | 0 | no |
| T | P28 | no | yes | no | no | 0 | no |
| T | P29 | no | yes | no | no | 0 | no |
| N | P30 | no | no | no | no | 0 | no |
| T | P31 | no | yes | no | no | 0 | no |
| N | P32 | no | no | no | no | 0 | no |
| N | P33 | no | no | no | no | 0 | no* |
| N | P34 | no | no | no | no | 0 | no |
| T | P35 | yes | yes | no | no | 0 | no |
| N | P36 | no | no | no | no | 0 | no |
| N | P37 | no | no | no | no | 0 | no* |
| T | P38 | yes | yes | no | no | 0 | no |
| N | P39 | no | no | no | no | 0 | no |
| N | P40 | no | no | no | no | 0 | no* |
| N | P41 | no | no | no | no | 0 | no |
| T | P42 | no | yes | no | no | 0 | no |
| N | P43 | no | no | no | no | 0 | no |
| P | P44 | yes | no | yes | yes | 5 | yes |

The group coding is the same as for Table 1.

CNAs; cancer-associated copy number alterations. CIN; chromosomally instable. WC; WISECONDOR (plasma sample). Ichor; ichorCNA (plasma).

a; a single amplification with Z-score 3–4, likely a false positive.

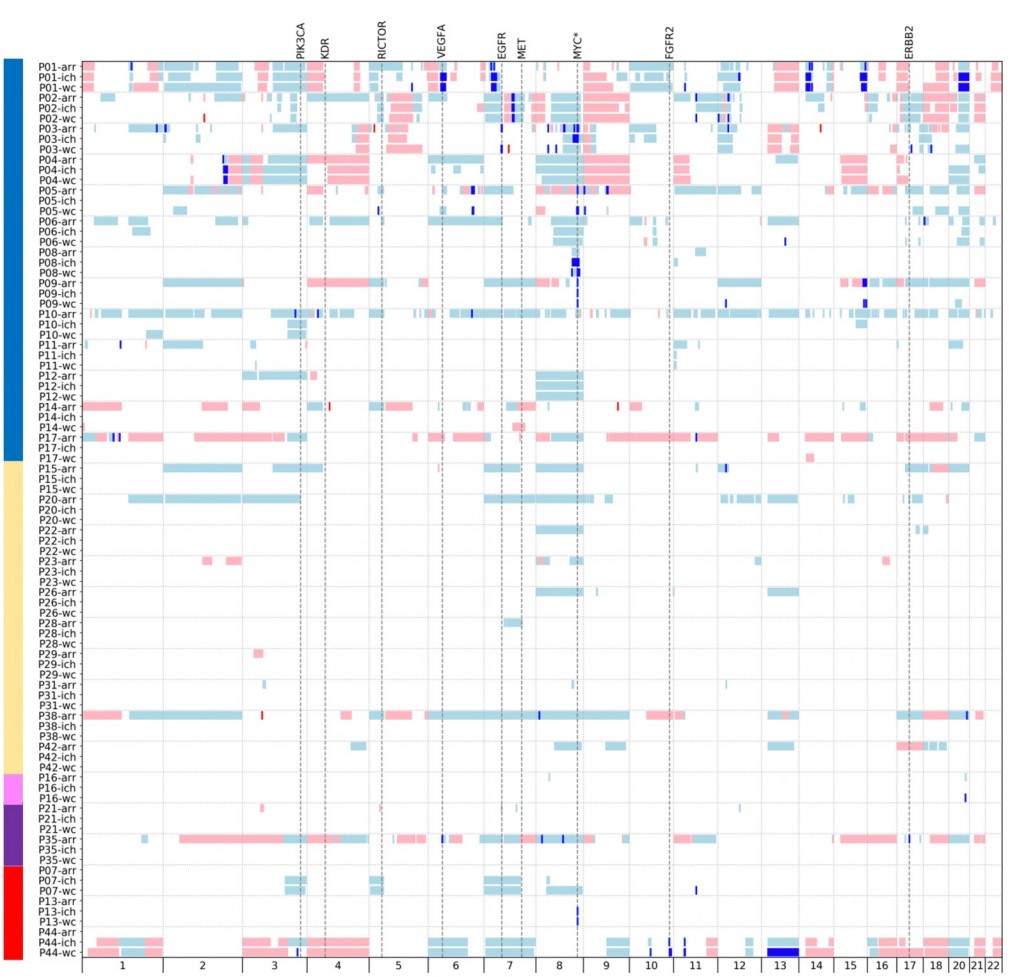

**Fig 2. Cancer-associated copy number alterations detected in plasma and paired tumour tissue biopsies.** All 29 patients with cancer-associated copy number alterations in tissue and/or plasma are included in the figure. Blue boxes denote gains and red denote losses. A dark blue shade indicates an amplification (gain with five or more copies) and dark red the equivalent negative ratio for a loss. Small gains or losses have been expanded slightly to enhance their visibility in the figure. Large calls spanning the centromere have been depicted as one CNA although the centromere region is uncallable. Chromosome 19, X and Y were excluded in the analysis and are therefore not shown in the figure. The genomic position of clinically potentially actionable regions for precision medicine (and the recurrently aberrant MYC gene, marked with *) have been indicated by vertical dotted lines in the figure. To the left are colour indicators referring to the subdivision of the patients as in Fig 1.

## CNAs detected in ctDNA from plasma

Targeted analysis of 61 regions known to harbour recurrent gains or losses in gastro-oesopha-geal cancer using WISECONDOR detected 18 CNAs with a z-score >5.0 in 11 patients, and 23 cancer-associated CNAs in 15 individuals if a cut-off of z-score 3.0 was used (Table 3).

In the four additional samples just passing the lower z-score threshold, there were four calls with a borderline z-score between 3.0 and 3.7 and none of these could be verified in the tissue analysis, suggesting that they were most likely false positives calls. In one, P03, a gain in the *MYC* gene with a z-score of 4.7 was found. This gain was confirmed in the tissue analysis and was interpreted to be a true cancer CNA. Eight of the patients had a CIN-profile in the corresponding tumour sample. In all, the detected CNAs included five amplifications of the *MYC*

**Table 3. Targeted analysis in plasma.** Targeted analysis in WISECONDOR data of 61 selected genomic regions and genes (reported in S2 Table). All regions with a Z-score 3.0 or higher and their corresponding findings in the whole-genome targeted plasma analyses and tissue array-CGH are presented. The "Ratio" column contains the normalized coverage difference in the targeted analysis, corresponding to effect size in the whole-genome analysis.

| Individual | Target region | Targeted analysis | | Whole genome analysis | | |
|---|---|---|---|---|---|---|
| | | Z-score | Ratio | WC | Ichor | Tissue |
| P01 | CD44 | 14 | 63% | Amp | Gain | 0 [a] |
| P01 | ZNF217 | 7.0 | 27% | Amp | Amp | Gain |
| P01 | 15q26.1 | 6.9 | 32% | Amp | Amp | Amp |
| P01 | VEGFA | 5.7 | 21% | Amp | Amp | Gain |
| P02 | CDK6 | 33 | 71% | Amp | Amp | Amp |
| P02 | CCND1 | 12 | 26% | Amp | Gain | Amp |
| P03 | EGFR | 160 | 439% | Amp | 0 | Amp |
| P03 | MYC | 4.7 | 11% | Gain | Amp | Amp |
| P04 | 2q32.1 | 8.6 | 21% | Amp | Amp | Amp |
| P05 | MYC | 28 | 50% | Amp | 0 | Amp |
| P06 | KLF5 | 8.1 | 16% | Amp | 0 | Gain |
| P07 | CCND1 | 12 | 24% | Amp | 0 | 0 |
| P08 | MYC | 79 | 114% | Amp | Amp | Gain |
| P09 | KRAS | 11 | 20% | Amp | 0 | Gain |
| P09 | MYC | 9.6 | 18% | Amp | Amp | Amp |
| P11 | 21q11.2 | 3.2 | 5% | 0 | 0 | 0 |
| P13 | MYC | 12 | 20% | Amp | Amp | 0 |
| P33 | FGFR2 | 3.2 | 6% | 0 | 0 | 0 |
| P37 | KDR | 3.3 | 6% | 0 | 0 | 0 |
| P40 | CD44 | 3.7 | 6% | 0 | 0 | 0 |
| P44 | FGFR2 | 124 | 382% | Amp | Amp | 0 |
| P44 | CD44 | 81 | 317% | Amp | Amp | 0 |
| P44 | KLF5 | 6.5 | 20% | Amp | Gain | 0 |

Amp; amplification, indicates an approximated gain of 5 or more copies. The ratio corresponds to the effect size and is the difference in the normalized sequencing coverage for the target bin/region. WC; WISECONDOR (whole-genome plasma analysis). ichor; ichorCNA (whole-genome plasma analysis). Tissue; array-CGH in tumour tissue biopsy.

[a]; The gain on chromosome 11 detected in tissue array-CGH does not include the *CD44* gene.

gene. In the tissue array-CGH analysis, three of the *MYC* gene amplifications were visible as amplifications and one as a gain. The fifth patient (P13) had tissue sampled after neoadjuvant chemotherapy and no CNAs were detected by array-CGH.

Initially in the whole-genome analysis, the output from WISECONDOR and ichorCNA comprised 449 and 234 CNAs with a minimum effect size of 1.5% and excluding the CNAs from chromosome 19, X and Y. In WISECONDOR, 166 CNAs in 17 patients (39% of the patients) were classified as cancer-associated (S5 Table). 133 CNAs in 14 patients were verified in the tissue array-CGH. In ichorCNA, 156 CNAs in 13 patients (30% of the patients) were classified as cancer-associated (S5 Table). 125 CNAs in 10 patients were verified in the tissue array-CGH. Among the 26 individuals with tumour tissue CNAs, the WISECONDOR plasma cancer detection rate in the whole-genome analysis of our cohort was 54% (14/26). S2 Fig shows the results in P04 from both whole-genome analyses software programs and the tissue array-CGH, as an example.

The whole-genome analysis detected plasma aberrations in all 11 patients that were previously detected by the target analysis using a Z-score of 5.0 as a threshold. In addition, six other

patients had CNAs detected by WISECONDOR using whole-genome analysis. ichorCNA detected 10 of the 11 found using the targeted analysis as well as an additional three patients in the whole-genome assay. All five of the *MYC* gene hits on chromosome 8q in the targeted analysis (patients P03, P05, P08, P09 and P13) were detected in the whole-genome analysis in plasma using WISECONDOR. However, one of them, in P03, was classified as a gain and not an amplification. In fact, chromosome 8 showed the largest number of CNAs: 21 in WISECONDOR (whereof 19 gains), and 15 in ichorCNA (whereof 13 gains) occurring in 10 patients. The highest *MYC* gene amplification was seen in P08, with a ctDNA fraction of 3.5% and an effect size of 88% corresponding to an estimated copy number of approximately 50.

There were in total 48 and 14 amplifications detected in WISECONDOR and ichorCNA, respectively. Potentially clinically actionable amplifications were detected in patients P01, P03 and P44 in the genes *VEGFA*, *EGFR*, and *FGFR2*.

In the total cohort of 44 patients, 15 CIN cases were detected. 14 of these were defined by their profile in the tissue array-CGH but patient P44 had cancer-associated CNAs present on more than 10 chromosomes only visible in plasma and not in the tissue array-CGH, which was sampled after neoadjuvant therapy. Out of the 15 patients with a CIN profile in either the plasma or tissue analysis, 11 (73%) had at least one cancer-associated CNA in plasma. In the non-CIN group, the corresponding number was 6/29 (21%).

Three patients had tumour-associated CNAs in plasma, which were not found in the analysis of tumour tissue. The tumour tissue sampling was in all these three patients performed after neoadjuvant chemotherapy. A clear *MYC* amplification was seen in individual P13 using both software, large WISECONDOR and ichorCNA CNAs with high effect sizes on chromosome 3, 5, 7 and 8 were seen in individual P07 and both software detected gains, amplifications and losses on several chromosomes in individual P44.

In the twelve samples where ichorCNA could estimate a ctDNA fraction, it was reported as 2.5–8% (Table 2). In two of the samples with an estimated ctDNA fraction 0%, ichorCNA did, however, find a tissue-verified *MYC* gene amplification (P09 and P13). There was no clear correlation between either ctDNA fraction or effect size and tumour size, tumour type, stage or distant metastasis in our small cohort. There was, though, a tendency but no significant correlation, for larger tumour sizes in the group with cancer-associated CNAs in plasma (S3 Fig).

In an effort to elucidate what might be a relevant threshold for cancer-associated CNAs even if tissue samples were not available, all CNAs from patients with tissue samples taken before neoadjuvant chemotherapy were analysed. The length of the aberration was plotted against the effect size (S4 Fig). In WISECONDOR, all CNAs >30Mb long and with an effect size >3% were verified cancer-associated CNAs. The same applies for all CNAs between 5 and 30Mb with an effect size >5%. For ichorCNA, a threshold of 30Mb in size and an effect size >3% or a size between 2 and 30Mb and an effect size of 5% identified verified cancer-associated CNAs with no unverified CNAs. In this group of patients, there were three CNAs in total that were classified as cancer-associated even though they were not initially verified in the tissue array-CGH. Two of them, both gains on chromosome 20 in P04, were detected in WISECONDOR and ichorCNA and upon re-evaluation of the array-CGH, a small deviation of the log2 ratio, suggesting a subclonal gain of a large part of chromosome 20, was detected but considered too low in effect size in comparison to the rest of the tissue analysis to confidently mark as cancer-associated. The third non-verified but cancer-associated plasma call was a loss detected on chromosome 7 in P14. The array-CGH showed a variable level of gains and losses in this region, including one gain spanning the first half of the plasma CNA and a loss in the second part, and therefore it could not be classified as a clear loss in the tissue analysis.

## Discussion

In this pilot study of 44 individuals with gastric and oesophageal cancer, we have shown that low-coverage whole genome sequencing of cell free DNA can detect CNAs in 39% (17/44) of all patients. Of the patients with known CNAs in tumour tissue 54% (14/26) had cancer-associated CNAs in plasma.

An advantage with this study is the use of a clinically validated workflow for non-invasive prenatal tests. Owing to this approach, the potential step from research to implementation in clinical routine is short. The use of a large reference set, handled in the same way as the cancer patient samples, and analysis of paired tumour tissue and plasma samples, enabled verification to reduce false positive CNAs in plasma from our cancer cohort. Since we used two different bioinformatic software programs in the same setting, we could compare their performance and evaluate their applicability to our clinical pipeline.

We started by using a targeted approach, including 61 regions in the plasma analyses. By using a z-score cut off of 3.0 we would expect 2–3 false positives (~1/1000 x 61 x 44) and we suspect that four of the 15 patients that had detectable CNAs using this approach were in fact false positives, since their CNAs were not verified in tissue or whole-genome analysis and three of them did not even have any cancer-associated CNAs detectable at all in any of the subsequent analyses of plasma and tissue. Using a more stringent z-score cut-off of 5.0 enabled detection of CNAs in 11 patients. By using a z-score limit of 4.0 we could identify one additional CNA in P03 (who already had another hit in the targeted analysis): a *MYC* gene amplification verified in both whole-genome plasma and tissue analysis.

Out of the 61 regions, only 23 contained CNAs with a z-score of 3.0 or higher, possibly suggesting that our region selection was not optimal. We based the panel of target regions primarily on the results from the Cancer Genome Atlas Research Network, including 295 gastric cancer samples [3]. 60% of these regions were replicated in the whole genome sequencing study on 168 treatment naïve gastric cancer samples in a Chinese study, suggesting that they are biologically relevant [21] and 75% of the peak regions of focal amplification presented by Schumacher et al [11] are included in our targeted region list. A subset are known recurrent amplifications of e.g. the *MYC*, *ERBB2* and *EGFR* genes [29]. Tumours are often aneuploid with multiple subclonal events and our analysis methods (array-CGH and whole-genome sequencing) can only measure the total ratio or coverage over each region (i.e. the sum of all the gains and losses) in relation to the mean in controls. In addition, gastro-oesophageal cancer is highly heterogeneous regarding CNAs with most recurrent gains and losses present in a minority of cases [11]. This complicates the interpretation of some regions.

To increase the detection rate, we expanded the plasma analysis in a whole-genome approach. This does not require prior knowledge of specific cancer aberrations. The whole-genome approach increased the number of individuals with cancer-associated CNAs detected in plasma to 17, as compared to 11 in the targeted analysis. In late stage cancers, whole-genome CNA analysis using low-coverage MPS on ctDNA may be a cost-effective measurement of disease burden and it is applicable across different cancer types when the ctDNA fractions are high enough [30, 31]. Since gastro-oesophageal cancer is a genetically complex disease but gains and losses are seen in more than half of the patients [11, 23, 32], our approach is suitable for ctDNA analysis even when a tumour tissue sample is not available.

Recurrent arm-level chromosome gains have previously been reported in more than 50% of gastric cancers, on chromosome 20q, 20p and 8q, and losses on chromosome 18q and 21q are seen in about 40% of the cases [11]. In the gastric cancer patients in our study, gains in tissue and/or plasma on chromosome 20q, 20p and/or 8q could be seen in 18 patients, corresponding to 78% (12 out of the 23 patients with gastric cancer and any detected CNA in plasma or

tissue). Losses on chromosome 18q and/or 21q could be seen in 10 patients (43%). In oesophageal adenocarcinomas, gains frequently occur on chromosomes 6q, 7p, 7q, 8q, 11q, 15q, and 17q and losses are common on chromosomes 1p, 3p, 4q, 8p, 9p, 17p, and 18q [33]. All three patients with oesophageal cancer in our study had CNAs overlapping these regions. Oesophageal squamous cell carcinomas have recurrent gains on chromosomes 7, 8 and 11 and losses on chromosomes 2, 3 and 7, for instance [32]. The two patients in our study with squamous cell carcinoma had gains overlapping those regions.

Clonal haematopoiesis is a common source of CNAs in plasma [34]. Clonal mosaic copy number gains in blood cells in healthy controls are recurrent at chromosome 8, 12 and 15 [35, 36] and can be misinterpreted as cancer-related. Of note, ~80% of the patients as well as the cancer-associated plasma CNAs detected in our study were also verified in the tissue biopsies. Therefore, we are confident that we have avoided most false positives due to clonal haematopoiesis in the plasma samples. However, if plasma-analysis is to be done without tumour sample analysis, it is recommended to compare the results from the ctDNA to those of DNA from peripheral blood in cases where that is needed in order to identify CNAs originating from leukocyte clonal haematopoiesis.

WISECONDOR provided a higher CNA detection rate and more amplifications compared to ichorCNA. This is in accordance with a previous report, arguing that the normalization process is a key step in the bioinformatic analysis [37]. WISECONDOR has an optimized normalization procedure that is based on a principal component analysis for the selection of a set of reference bins, usually ~100 bins across many chromosomes for each target bin. In the version of WISECONDOR applied in this study, gains are not always correctly segmented and smaller amplifications may therefore be masked by larger overlapping or nearby gains. For example, in the tissue array-CGH and the whole-genome analysis with ichorCNA from P03, an amplification of the *MYC* gene was detected. In WISECONDOR, only a gain was detected in the region. In a further development of the original version, (WisecondorX), segmentation has been altered to address this potential problem [37]. Also, in contrast to WISECONDOR, ichorCNA does provide a calculation of the ctDNA fraction, which can be an advantage, for instance in the interpretation of gains versus amplifications.

Amplification in a few genes are considered potentially actionable in gastric cancer [2, 23–27]. In total, four patients had such amplifications and all four of them had a CIN tumour profile. In both P01 with plasma and tissue sampled before chemotherapy, amplifications (in *VEGFA* and *CD44)* detected in plasma were called as gains in the tissue by array-CGH. One possible explanation for this discrepancy is tumour heterogeneity. Another example of potential tumour heterogeneity can be seen in P03 (Fig 2), who also had an amplification of *EGFR* detected in both plasma and tissue. P35 had amplifications of the *VEGFA* and *ERBB2* genes in tissue only, but the plasma sample was taken after chemotherapy. P44 had an amplification in the *FGFR2* gene in plasma only and the post-treatment tissue biopsy was negative despite a clinical stage IV, showing the difficulties in capturing relevant cells in small tissue biopsies. *ERBB2* amplification analysis, mostly by immunohistochemistry for ERBB2 expression, is currently being introduced as a standard clinical test in gastro-oesophageal cancer patients in the clinic, in particular in a metastatic situation, but was not yet standard procedure at the time of the tissue sample collection for this study. Therefore, only two patients (P21 and P35) had a clinically detected *ERBB2* overexpression. In P21, no *ERBB2* amplification could be seen in plasma or the tissue array-CGH (which did contain other detectable CNAs), but in P35, the *ERBB2* amplification was detected in tissue (Fig 2).

According to the ichor-CNA developers, their algorithm can reliably detect a ctDNA fraction of at least 3%. In addition, at least one amplification and one deletion event, both larger than 100Mb are needed in order to provide an accurate estimation of ctDNA fraction [16, 38].

In all, six samples with detectable CNAs in plasma had no estimation of ctDNA fraction. Five harboured only 1 or 2 small CNAs making an estimation of ctDNA fraction by ichor-CNA impossible (P09, P13, P14, P16 and P17). Only one sample (P05) had several larger CNAs called by WISECONDOR, but these were not detected by ichorCNA, thus the lack of a ctDNA fraction estimation is likely due to a low ctDNA fraction.

The sensitivity of low coverage whole genome sequencing for CNAs in cell free DNA is dependent on both the technical sensitivity of the method and the biological variation in the samples. The biological sensitivity depends on whether or not the tumour had CNAs at all, as well as on its propensity to shed cell free DNA into the circulation and thus yield a higher ctDNA fraction in plasma. In addition, subclonal events will be more difficult to detect than early CNAs that are present in all or a large majority of tumour cells. In many cases, the ctDNA fraction will be the determining factor for the technical sensitivity. For prenatal screening, a minimum fraction of foetal DNA in the total cell free DNA of 2.0% is required in order to detect trisomy 13, 18 or 21 [39] but there are differences between different platforms and some require at least 4%. Of note, most studies that use low-coverage whole genome sequencing require a ctDNA fraction of at least 5–10% [16, 40]. Among our positive prenatal samples, all foetal trisomies were detected by both WISECONDOR and ichorCNA and the lowest foetal fraction was 5% (S3 Table). Among the cfDNA samples from cancer patients, WISECONDOR and ichor-CNA could detect CNAs in samples with a ctDNA fraction of 0–8.4%, and 7/11 (64%) had a ctDNA fraction of less than 5% (Fig 1). This is in line with other studies; the ctDNA fraction in advanced gastric cancer was reported to be 0.3–8% with a median of approximately 1.6% [41] and in early cancer stages of other gastrointestinal cancers the fractions are even lower (0–0.8%) [42].

Our analysis of tumour biopsies taken before neoadjuvant chemotherapy showed that 23/30 (77%) tumours harboured CNAs, thus we would expect at most 23 samples to have positive ctDNA results. Of these, 13 had positive ctDNA results and ten were negative. Four of the negative samples (P23, P28, P29, P31) had CNAs less than 50Mb in size with a small effect size on array-CGH analysis, which were likely below the technical detection limit of the algorithms. Six samples had large CNAs with at least one trisomy (P15, P20, P22, P26, P38, P42) (Fig 2). The CNAs found in tissue in P15 had a low effect size (S4 Table) and were likely subclonal events that were under the threshold of detection in plasma. The other five samples had a ctDNA fraction of 0 (i.e. were tumours that did not shed much ctDNA) and thus their ctDNA was negative. It is not yet established which factors are the most important for determining the level of ctDNA fraction, although localization, size, stage, kidney clearance, age, invasiveness have all been suggested [43]. Our cohort is too small to robustly analyse these parameters, but we did see a tendency towards a correlation between larger tumour size and higher ctDNA levels. We used 1 ml of plasma from each individual, according to the standard NIPT protocol. Increasing the sample input volume to 3 ml plasma does not increase the detection of samples with low ctDNA fraction [44], but will only increase the cost [16].

To date, there are very few studies on CNAs in ctDNA in gastro-oesophageal cancer cohorts, making this study an important contribution. Most reports on plasma CNA detection are small proof-of-concept studies [45] and the majority of the studies investigating ctDNA in gastro-oesophageal cancer include predominantly individuals of Asian ancestry [31]. WISECONDOR or ichorCNA have been used for plasma analyses of CNAs in ctDNA in cancer. Cohen et al report on the application of WISECONDOR in a cohort of 32 women with ovarian cancer and 32 benign controls [46]. Adalsteinsson et al used ichorCNA in a cohort of 520 patients with metastatic prostate and breast cancer, comparing to tissue analyses in 41 patients [16].

One study by Davidson et al used ichorCNA on a cohort of 30 individuals with gastro-oeso-phageal adenocarcinoma with no reference set [44]. CNAs were detected in 23 (77%) of the individuals using a whole-genome approach and ichorCNA bioinformatics, with recurrent gains on for instance 8p. The approximate same region on 8p also showed gains in 6 of the patients in our cohort. In a targeted approach with 50kb bins, they found tissue-verified ampli-fications. Their study cohort included only advanced inoperable (10%) or metastatic (90%) cancers, while only 16% of our cohort comprised patients with verified metastases. Also, 80% of the patients included in the study by Davidson et al were cancers in the oesophagus, while only 5 (11%) in our cohort were, which might explain the differences in the detection rate. In addition, all CNAs in our cohort were filtered against a reference set with exclusion of 63% of all calls from WISECONDOR and 34% of all CNA calls from ichor. Therefore, our data are not immediately comparable to the data presented by Davidson et al. Another study, using whole-genome low coverage analysis with a focus on chromosomal instability scores in gastro-oesophageal cancer, identified 27/55 (49%) of patients with CNAs in plasma after comparing to DNA from peripheral blood cells [47] and Maron et al analysed a targeted panel on ctDNA including amplifications with a detection of multiple amplifications in 40% of the gastro-oeso-phageal cancer patients [29], both in line with our results.

Different approaches of plasma DNA examination in patients with gastro-oesophageal can-cer have been reported, initially mostly analyses of total cell free DNA concentration, which is however, a non-specific test for cancer [12]. Another approach for ctDNA analysis in gastro-oesophageal cancer is single nucleotide variant analysis. Most of the cohorts are small and diverse when it comes to tumour subtype and stage, with a detection rate spanning 20–80%. The technical approach also differs between studies, using either a personalised panel adapted to known tumour tissue genetic aberrations from the same individual [48–52] or a pre-set panel [29, 53–59].

It is well known that tumour tissue biopsies have limitations. Naturally, sometimes the tumour location makes a tissue biopsy procedure difficult. In addition, due to tumour hetero-geneity, a single tumour biopsy may only represent a small clone that is not representative of the major tumour burden. In fact, before using ERBB2-inhibitor therapy in gastro-oesophageal cancer, at least five biopsies are recommended in order to ensure reliable results [7]. Thus, ctDNA analysis might provide more comprehensive, or additional CNA information [60]. An example is the amplification of the *VEGFA* gene in P01, which is only visible as a gain in the tissue-array-CGH but a clear amplification in the plasma analyses.

All newly diagnosed patients with gastroesophageal cancer that are potentially operable are referred to the Department of Upper Abdominal Diseases at Karolinska University Hospital. Most of the patients who were eligible for participation in the study accepted. The gender (30% women) and age (median 70 years) of the patients included in this study are comparable to the 40% women and median age 72 years reported for gastric cancer in Sweden 2018 [61]. The most common tumour stage in Stockholm 2016–2018 was III [62], and that was also the most common stage of the patients included in this study. The study cohort is thus representa-tive of the population with gastroesophageal cancer in Stockholm county.

Our study reflects the clinical situation, where patients often perform their diagnostic biopsy in another medical centre before being referred to the university hospital for treatment and fresh frozen tumour tissue is not always available before initiation of treatment. Drawing a blood sample with analysis of plasma DNA with rigorous filtering might in these cases provide important information without the need for a second gastroscopy. Of note, two of the partici-pants had squamous cell cancer in the oesophagus. It is known that the genetics of squamous cell tumours and adenocarcinomas differ and more studies on ctDNA in both these groups are needed to be able to know if the same approach can be used in liquid biopsy for both groups.

The clinical stage of the cancer in our study was estimated in a multidisciplinary tumour board consisting of experienced gastro-oesophageal cancer surgeons, oncologists, radiologists, pathologists and endoscopists and was based on gastroscopy, biopsy and CT/PET-CT (computer tomography, positron-emission tomography) scans. The complete histopathological report on the resected tumour specimen, received after surgery, can be more accurate, but has the disadvantage of often being made after neoadjuvant chemo- or chemoradiation treatment and in many patients no surgery is performed. Therefore, although there was no correlation between the clinical stage of the patients and the ctDNA fraction estimate or the effect size in our study, such a correlation cannot be ruled out and should be addressed in a larger cohort.

## Conclusions

In summary, low-coverage whole-genome sequencing without prior knowledge of the tumour aberrations is a useful tool for ctDNA analysis of total copy number alterations in plasma from patients with gastro-oesophageal cancer. It can detect chromosomal instability as well as clinically actionable amplifications in genes important for therapy such as *ERBB2* and *EGFR* and is thus an important complement to more traditional gene panel analyses that target single nucleotide variants. In addition, liquid biopsies are minimally invasive and provide overall information on the genetic aberrations regardless of tumour heterogeneity. Further studies are needed on longitudinal liquid biopsy samples from more gastro-oesophageal cancer patients in order to follow tumour dynamics and further investigate the sensitivity of the method.

## Supporting information

**S1 Fig. Sample bin coverage variation in relation to number of calls.** Mean absolute error of the normalized coverage difference between all bins and the number of calls for all of the samples in the reference set (n = 414). The mean absolute error is plotted on the X cropped axis and the number of copy number alterations called by WISECONDOR (S1a) and ichorCNA (S1b) are plotted on the Y axis.
(TIF)

**S2 Fig. Example of the analysis results from tissue and plasma for patient four.** In the tissue array-CGH (S2a) copy number alterations (CNAs) after filtering are indicated by blue boxes with gains above the zero line and losses below. In the upper panel are the general overview chromosomal positions and, on the Y-axis, the log2 ratio of each probe is shown as dots. The moving average is indicated by a blue line. In the low-coverage whole-genome analysis in plasma using WISECONDOR (S2b), the blue line indicates the bin Z-score. Called regions (before filtering) are indicated by yellow/green boxes depending on the effect size together with the Z-score for the region. In the low-coverage whole-genome analysis in plasma using ichorCNA (S2c) dots represent bins with their log2 ratio shown on the Y-axis. Regions with gains, including amplifications, are indicated by brown/red colour and losses are indicated by green.
(TIF)

**S3 Fig. Tumor size versus CNA detection in plasma.** Tumour size [cm] in the group with detectable CNAs in plasma (blue box) and in the group with no detectable CNAs in plasma (orange box). Mann-Whitney, plotted in R software. p = 0.06056.
(TIF)

**S4 Fig. Effect size in relation to the length of aberration for verified and unverified plasma copy number alterations (CNAs) in patients with gastro-oesophageal cancer.** Plot of all cancer-associated CNAs in plasma from individuals (n = 13) with tissue samples taken before any

chemotherapy and CNAs called by WISECONDOR (S4a) and ichor (S4b). On the log10 Y-axis, effect size of the CNA and on the X-axis, size in megabasepairs (Mbps) of the CNA. CNAs verified in the tissue array-CGH are coloured orange and the "unverified" are coloured blue. The CNAs coloured grey (n = 3 in total) were manually classified as cancer-associated even though they were not visible in the tissue array-CGH (see Methods for details).
(TIF)

**S1 Table. Regions and genes in the targeted plasma analysis.** All 61 regions included in the targeted analysis. Cytoband and refseq gene positions are provided in genome build GRCh37/hg19.
(DOCX)

**S2 Table. WISECONDOR and ichorCNA calls in the reference samples.** All gains and losses called in the reference set (n = 414) with the corresponding frequency. Positions are provided in genome build GRCh37/hg19.
(XLSX)

**S3 Table. Foetal positive control NIPT samples.** NIPT analysis results of plasma from 15 pregnant women carrying foetuses with trisomy 21, trisomy 18 or trisomy 13 (verified by invasive test). All trisomies were correctly detected by WISECONDOR and ichorCNA. For each sample the foetal trisomy chromosome and the estimated foetal fraction are provided in separate columns.
(DOCX)

**S4 Table. Cancer-associated copy number alterations in tumour tissue.** All cancer-associated CNAs detected by array-CGH analysis in the tumour tissue analysis of the 44 patients. Positions are provided in genome build GRCh37/hg19.
(XLSX)

**S5 Table. Cancer-associated copy number alterations in plasma.** All cancer-associated CNAs detected by WISECONDOR and ichorCNA in the plasma from the 44 patients. Positions are provided in genome build GRCh37/hg19.
(XLSX)

**S6 Table. CNAs in P03 detected by gene panel analysis.** Results from the gene panel comprising 370 genes including *EGFR* (chromosome 7) and *ERBB2* (chromosome 17) performed on tumour DNA from P03. The amplification of EGFR was confirmed and the amplification onchromsome 17 comprised the genes: *STAT3, CNTNAP1, EZH1, AOC3, RND2, BRCA1*, but not *ERBB2*.
(XLSX)

## Acknowledgments

First of all, we would like to thank all patients participating in the study.

We would also like to thank the staff at The Department of Clinical Genetics, Karolinska University Hospital Solna without whom this study would not have been possible. Also, Valtteri Wirta and his staff at Clinical Genomics, SciLifeLab, Stockholm for their professional contributions. We thank Berit Sunde for her commitment in informing patients about the study and collecting samples and we thank Ollanta Cuba Gyllensten for creating the figure outputs and helping with technical issues. Some of the computations were performed using SNIC through Uppsala Multidisciplinary Center for Advanced Computational Science (UPPMAX).

## Author Contributions

**Conceptualization:** Karin Wallander, Mats Lindblad, Magnus Nordenskjöld, Agne Liedén, Emma Tham.

**Data curation:** Daniel Nilsson, Kenny Billiau, Hassan Foroughi.

**Formal analysis:** Karin Wallander, Jesper Eisfeldt, Agne Liedén.

**Funding acquisition:** Karin Wallander, Magnus Nordenskjöld.

**Investigation:** Karin Wallander, Agne Liedén, Emma Tham.

**Methodology:** Karin Wallander, Jesper Eisfeldt, Agne Liedén, Emma Tham.

**Project administration:** Emma Tham.

**Resources:** Mats Lindblad, Kenny Billiau.

**Supervision:** Magnus Nordenskjöld, Emma Tham.

**Visualization:** Karin Wallander, Agne Liedén.

**Writing – original draft:** Karin Wallander, Agne Liedén, Emma Tham.

**Writing – review & editing:** Jesper Eisfeldt, Mats Lindblad, Daniel Nilsson, Hassan Foroughi, Magnus Nordenskjöld.

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
