## [Decision Letter · Decision Letter 0]

21 Aug 2020

PONE-D-20-14603

Cell free tumour DNA analysis detects copy number alterations in gastro-oesophageal cancer patients.

PLOS ONE

Dear Dr. Tham,

Thank you for submitting your manuscript to PLOS ONE. After careful consideration, we feel that it has merit but does not fully meet PLOS ONE’s publication criteria as it currently stands. Therefore, we invite you to submit a revised version of the manuscript that addresses the points raised during the review process.

I typically want more than one reviewer; however, to expediate the process I am going with this single review.  Please attempt to address the reviewers concerns. 

We look forward to receiving your revised manuscript.

Kind regards,

Jeffrey Chalmers, Ph.D.

Academic Editor

PLOS ONE

Journal Requirements:

2. Please provide additional details regarding participant consent.

In the ethics statement in the Methods and online submission information, please ensure that you have specified whether consent was informed.

3. Please note that PLOS does not permit references to “data not shown.” Authors should provide the relevant data within the manuscript, the Supporting Information files, or in a public repository.

Please add a citation to support this phrase or upload the data that corresponds with these findings to a stable repository (such as Figshare or Dryad) and provide and URLs, DOIs, or accession numbers that may be used to access these data.

If the data are not a core part of the research study being presented, we ask that authors remove any references to these data.

4. In your Methods section, please provide additional information about the participant recruitment method and the demographic details of your participants. Please ensure you have provided sufficient details to replicate the analyses such as: a) a table of relevant demographic details and b) a statement as to whether your sample can be considered representative of a larger population.

6. Your ethics statement must appear in the Methods section of your manuscript. If your ethics statement is written in any section besides the Methods, please move it to the Methods section and delete it from any other section. Please also ensure that your ethics statement is included in your manuscript, as the ethics section of your online submission will not be published alongside your manuscript.

Reviewers' comments:

Reviewer's Responses to Questions

**Comments to the Author**

1. Is the manuscript technically sound, and do the data support the conclusions?

Reviewer #1: Yes

2. Has the statistical analysis been performed appropriately and rigorously? 

Reviewer #1: Yes

3. Have the authors made all data underlying the findings in their manuscript fully available?

Reviewer #1: Yes

4. Is the manuscript presented in an intelligible fashion and written in standard English?

Reviewer #1: Yes

5. Review Comments to the Author

Reviewer #1: Review of Wallander, et al.

“Cell-free tumour DNA analysis detects copy number alterations in gastro-oesophageal cancer patients.”

In this manuscript, the authors investigate copy number alterations in gastro-esophageal cancer tumor samples (via aCGH) and paired plasma (via shallow whole genome seq). They compare two algorithms for plasma ctDNA analysis.

The strengths of this manuscript include their analysis of tumor and ctDNA as well as comparison of methodologies.

This manuscript could be improved through several items, detailed below. The biggest challenges I see is the lack of discussion about that *would* be expected to be detectable based on tumor DNA in plasma. The analysis is largely descriptive but hard to interpret the analysis and how the authors see the potential implementation. I believe this can be addressed through further analysis of existing data, plus some additional validation of tumor samples.

MAJOR COMMENTS

-Comparing tumor (aCGH) with plasma ctDNA: A major challenge is that analyses of copy number were different in tumor (aCGH) and ctDNA (sWGS; WISECONDOR and ichorCNA). Did the authors consider performing shallow WGS on the tumor samples (or a subset of samples)? If DNA was extracted for aCGH, could be sequenced as the plasma samples (but with higher tumor content), then could use the same algorithms across all samples. This would be a nice validation to show that it was not the aCGH that caused variation – which the authors cite

-ctDNA tumor content: The authors make little mention of whether we would EXPECT to find CNAs in plasma. Across cancers, some patient tumors ‘shed’ while others don’t. if there is a non-shed tumor we would not anticipate to find CNAs in plasma. It appears that the agreement in detection of CNAs was similar between WISECONDOR/ichorCNA suggesting that many of the samples where CNAs were not detected are just ‘non-shed’ tumors. This is exemplified by the number of patients where ctDNA fraction was 0. Could the authors revisit whether we would even expect to see these in plasma?

-Copy number detection threshold: Could the authors discuss further the anticipated lower limit of detection for copy number alternations? For example, would we expect WISECONDOR and ichorCNA to detect copy number alterats and ctDNA fraction of 1%?

MINOR COMMENTS

-

6. PLOS authors have the option to publish the peer review history of their article (what does this mean?). If published, this will include your full peer review and any attached files.

Reviewer #1: No

---

## [Author Response · Author response to Decision Letter 0]

6 Oct 2020

Response to reviewers

Please see our responses below each point.

1. Please ensure that your manuscript meets PLOS ONE's style requirements, including those for file naming. The PLOS ONE style templates can be found at (…)

We have updated the fonts, tables and figure formatting, and file names. 

2. Please provide additional details regarding participant consent.

In the ethics statement in the Methods and online submission information, please ensure that you have specified whether consent was informed.

All participants gave their written informed consent, as can be seen on line 161-164 in the Methods section. 

3. Please note that PLOS does not permit references to “data not shown.” Authors should provide the relevant data within the manuscript, the Supporting Information files, or in a public repository.

Please add a citation to support this phrase or upload the data that corresponds with these findings to a stable repository (such as Figshare or Dryad) and provide and URLs, DOIs, or accession numbers that may be used to access these data.

If the data are not a core part of the research study being presented, we ask that authors remove any references to these data.

There were four instances where we had stated “data not shown”. 

- We deleted the reference to “data not shown” on line 223-224, since it was not necessary (no results to show) 

- On line 311 and 413 we have added a supplementary table (S5) and figure (S3), respectively 

- The information on the reference set should have been in the methods section as part of our optimization protocol and thus there is no data to show. We have therefore moved the information on the reference set from line 370 to line 267-268 in the Methods section, and deleted the data reference.

4. In your Methods section, please provide additional information about the participant recruitment method and the demographic details of your participants. Please ensure you have provided sufficient details to replicate the analyses such as: a) a table of relevant demographic details and b) a statement as to whether your sample can be considered representative of a larger population.

We have provided additional information about the participant recruitment method and the demographic details are now presented in a separate table (Table 1). We have also added results from the publicly available Swedish cancer registry for comparison in the discussion. We have added a statement in the discussion comparing these two results that show that our study population is representative of the gastric cancer population of Stockholm County (see Discussion, line 654-662). 

We have provided a separate caption for each figure.

6. Your ethics statement must appear in the Methods section of your manuscript. If your ethics statement is written in any section besides the Methods, please move it to the Methods section and delete it from any other section. Please also ensure that your ethics statement is included in your manuscript, as the ethics section of your online submission will not be published alongside your manuscript.

We have moved our ethics statement to lines 161-164 (right after the selection of patients was described). Our study was approved by the Swedish Ethical Review Authority and all participants received oral and written information and provided their written consent.

Reviewer #1: Review of Wallander, et al.

“Cell-free tumour DNA analysis detects copy number alterations in gastro-oesophageal cancer patients.”

In this manuscript, the authors investigate copy number alterations in gastro-esophageal cancer tumor samples (via aCGH) and paired plasma (via shallow whole genome seq). They compare two algorithms for plasma ctDNA analysis.

The strengths of this manuscript include their analysis of tumor and ctDNA as well as comparison of methodologies.

This manuscript could be improved through several items, detailed below. The biggest challenges I see is the lack of discussion about that *would* be expected to be detectable based on tumor DNA in plasma. The analysis is largely descriptive but hard to interpret the analysis and how the authors see the potential implementation. I believe this can be addressed through further analysis of existing data, plus some additional validation of tumor samples.

MAJOR COMMENTS

-Comparing tumor (aCGH) with plasma ctDNA: A major challenge is that analyses of copy number were different in tumor (aCGH) and ctDNA (sWGS; WISECONDOR and ichorCNA). Did the authors consider performing shallow WGS on the tumor samples (or a subset of samples)? If DNA was extracted for aCGH, could be sequenced as the plasma samples (but with higher tumor content), then could use the same algorithms across all samples. This would be a nice validation to show that it was not the aCGH that caused variation – which the authors cite.

The reviewer is correct that analysis with shallow WGS on tumor material would likely be more sensitive than array-CGH. The problem is that in order to call tumor CNAs in the best way using the tested methods, you need a large set of normal reference tissue samples (100-200 samples), which have been processed the same way as the patient samples. We do not have such a reference set and have therefore not chosen shallow-WGS to detect CNAs in the tumour tissue. We have used array-CGH for many years in both the clinical and research setting and thus have experience to interpret the results and we have access to a large database of samples (~8 000) for comparison. Array-CGH would typically allow detection of copy number changes that occur in at least 10-20% of the analyzed cells, with higher sensitivity for large events (several megabases) and lower for smaller events (~20 kb – 1 Mb), but also depending on genomic localization.

Furthermore, although shallow WGS is more sensitive than array-CGH and could potentially detect more subclonal CNAs in tumor material, we believe that these subclonal events in most cases would not be detectable in cell-free DNA, see new part of discussion lines 574-576. Also, the largest contribution to differences between tissue and plasma analysis are likely tumour heterogeneity. Thus, the benefit of using a more sensitive method on the tumor DNA is perhaps questionable. 

-ctDNA tumor content: The authors make little mention of whether we would EXPECT to find CNAs in plasma. Across cancers, some patient tumors ‘shed’ while others don’t. if there is a non-shed tumor we would not anticipate to find CNAs in plasma. It appears that the agreement in detection of CNAs was similar between WISECONDOR/ichorCNA suggesting that many of the samples where CNAs were not detected are just ‘non-shed’ tumors. This is exemplified by the number of patients where ctDNA fraction was 0. Could the authors revisit whether we would even expect to see these in plasma?

We have rewritten this section of the Discussion (see lines 561-604), to clarify what we would expect to find in our results. 23 samples displayed CNAs in tumour tissue taken before any therapy. Of these, 13 had positive ctDNA results and 10 were negative. Of the 10 negative samples, four had small CNAs with low effect size that we would not expect to be detectable in plasma. Six had trisomies and larger aberrations in the tumour tisse, but were either subclonal or had too low ctDNA fractions (i.e. were “non-shedders”) and thus their plasma analyses were negative. 

-Copy number detection threshold: Could the authors discuss further the anticipated lower limit of detection for copy number alternations? For example, would we expect WISECONDOR and ichorCNA to detect copy number alterats and ctDNA fraction of 1%?

We have clarified this in the discussion, lines 561- 604, and above. 

Most studies require a ctDNA fraction of at least 5% in order to reliably perform low coverage WGS on plasma samples. For large alterations such as whole chromosome gains e.g. trisomy 21, a foetal/ctDNA fraction of at least 2.5% is required. Lower levels will not be detectable due to the normal variation in coverage among samples. By optimizing the analysis using a large panel of reference samples and rigourous curation, we have been able to reliably detect CNAs in plasma samples with a ctDNA fraction of 2.5 – 8%, where 7 of 11 samples had a ctDNA fraction below 5%. At lower ctDNA fractions, larger CNAs and/or larger effect sizes (i.e. amplifications) can be detected, but smaller CNAs and/or duplications will be missed.

---

## [Decision Letter · Decision Letter 1]

26 Oct 2020

PONE-D-20-14603R1

Cell free tumour DNA analysis detects copy number alterations in gastro-oesophageal cancer patients.

PLOS ONE

Dear Dr. Tham,

Thank you for submitting your manuscript to PLOS ONE. After careful consideration, we feel that it has merit but does not fully meet PLOS ONE’s publication criteria as it currently stands. Therefore, we invite you to submit a revised version of the manuscript that addresses the points raised during the review process.

As the reviewer commented, the authors really did not address the issued raised after the first review.  Please address the raised issues if you wish to have the paper published.

We look forward to receiving your revised manuscript.

Kind regards,

Jeffrey Chalmers, Ph.D.

Academic Editor

PLOS ONE

Reviewers' comments:

Reviewer's Responses to Questions

**Comments to the Author**

1. If the authors have adequately addressed your comments raised in a previous round of review and you feel that this manuscript is now acceptable for publication, you may indicate that here to bypass the “Comments to the Author” section, enter your conflict of interest statement in the “Confidential to Editor” section, and submit your "Accept" recommendation.

Reviewer #1: (No Response)

2. Is the manuscript technically sound, and do the data support the conclusions?

Reviewer #1: Yes

3. Has the statistical analysis been performed appropriately and rigorously? 

Reviewer #1: Yes

4. Have the authors made all data underlying the findings in their manuscript fully available?

Reviewer #1: Yes

5. Is the manuscript presented in an intelligible fashion and written in standard English?

Reviewer #1: Yes

6. Review Comments to the Author

Reviewer #1: While the authors provided responses to the critiques, they essentially changed nothing of substance in the manuscript. The focus of this manuscript is around ability to detect copy number alterations. The authors primary conclusions around detectability of copy number alterations in plasma versus tissue are completely reliant on whether these would be expected to be detected or not. The authors made a very marginal effort to address the queries, with no additional analysis and one revised paragraph in the discussion. If not possible to analyze this dataset, perhaps there is another? Even within the text, no adjustments to major conclusions (e.g. abstract) were made. All other revisions were editorial.

My suggestions to the authors remain unchanged as no additional analyses or interpretation has effectively been performed.

7. PLOS authors have the option to publish the peer review history of their article (what does this mean?). If published, this will include your full peer review and any attached files.

Reviewer #1: No

---

## [Decision Letter · Decision Letter 2]

2 Jan 2021

Cell free tumour DNA analysis detects copy number alterations in gastro-oesophageal cancer patients.

PONE-D-20-14603R2

Dear Dr. Tham,

We’re pleased to inform you that your manuscript has been judged scientifically suitable for publication and will be formally accepted for publication once it meets all outstanding technical requirements.

Kind regards,

Jeffrey Chalmers, Ph.D.

Academic Editor

PLOS ONE

Additional Editor Comments (optional):

Reviewers' comments:

Reviewer's Responses to Questions

**Comments to the Author**

1. If the authors have adequately addressed your comments raised in a previous round of review and you feel that this manuscript is now acceptable for publication, you may indicate that here to bypass the “Comments to the Author” section, enter your conflict of interest statement in the “Confidential to Editor” section, and submit your "Accept" recommendation.

Reviewer #1: All comments have been addressed

2. Is the manuscript technically sound, and do the data support the conclusions?

Reviewer #1: Yes

3. Has the statistical analysis been performed appropriately and rigorously? 

Reviewer #1: Yes

4. Have the authors made all data underlying the findings in their manuscript fully available?

Reviewer #1: No

5. Is the manuscript presented in an intelligible fashion and written in standard English?

Reviewer #1: Yes

6. Review Comments to the Author

Reviewer #1: With this revision, the authors have made a reasonable attempt to address my concerns. I have no additional concerns that can be reasonably addressed. The implications of the research remain not entirely clear but as this is a proof of concept study, would be beyond scope.

7. PLOS authors have the option to publish the peer review history of their article (what does this mean?). If published, this will include your full peer review and any attached files.

Reviewer #1: No

---

## [Editor Report · Acceptance letter]

6 Jan 2021

PONE-D-20-14603R2 

Cell-free tumour DNA analysis detects copy number alterations in gastro-oesophageal cancer patients.  

Dear Dr. Tham:

I'm pleased to inform you that your manuscript has been deemed suitable for publication in PLOS ONE. Congratulations! Your manuscript is now with our production department. 

Kind regards, 

on behalf of

Dr. Jeffrey Chalmers 

Academic Editor

PLOS ONE